# PRESERVING LARGE ACTIVATIONS: THE KEY TO KV CACHE PRUNING

## ABSTRACT

As context lenghts grows, the increasing size of Key and Value (KV) cache poses a significant challenge to efficiently serving Large Language Models (LLMs). KV cache pruning, by preserving only a small subset of important KV cache for sparse inference, is a recognized effective solution. Our research revealed that large activations are the key to identifying these important KV cache. However, existing methods have not been successful in effectively identifying these important KV cache due to neglecting the impact of Value cache, and are also incompatible with Grouped-Query Attention (GQA) architectures. To address these issues, we introduce an innovative KV cache pruning method that preserves these large activations and is compatible with Grouped-Query Attention. Featuring a novel pruning metric, this method operates within each attention group to enhance efficiency and minimize performance degradation. Experimental results demonstrate that our approach not only maintains comparable accuracy with existing methods but also significantly reduces KV cache requirements. Specifically, It demonstrates similar accuracy while utilizing only 1/10 of the KV cache compared to existing SOTA methods.

## 1 INTRODUCTION

Large Language Models (LLMs) have garnered significant attention due to their remarkable performance across various applications, such as ChatGPT (OpenAI, 2022), Claude (Anthropic, 2024) and Gemini (Team, 2024). However, as the context lengths utilized by these models increase, serving them efficiently becomes an increasingly challenging task. For example, the Llama-2-7B model (Touvron et al., 2023), when configured with a batch size of 16 to handle a context length of 32K, requires 256 GiB of VRAM solely for storing its KV cache and an additional 28 GiB for model weights. Given that the most advanced GPUs currently available offer a maximum of 80 GiB of VRAM, this results in substantial inference costs due to the latency involved in accessing large amounts of KV cache and the excessive consumption of valuable storage resources.

Pruning KV cache is a pivotal approach to efficient LLM serving. Specifically, during the prefill phase, LLMs generate extensive KV cache, and existing pruning methods utilize various strategies to assess the importance of KV cache, subsequently evicting the majority and retaining only an important subset for sparse inference computations during the generation phase. Our analysis has revealed that the key to KV cache pruning is preserving large activations. Recent studies (Sun et al., 2024) have indicated that large activations have a significant impact on model inference results and remove these large activations will lead to severe performance degradation. However, current methods do not adequately identify the important KV cache because they focus only on large activations within Key cache, neglecting those in the Value cache, leading to notable performance degradation. Our experiments indicate that large activations in the Value cache also have a significant impact on inference results. Furthermore, existing pruning methods are also incompatible with mainstream models employing more efficient Grouped-Query Attention (Ainslie et al., 2023) architectures, such as Llama-3 (Dubey et al., 2024) and Mistral (Jiang et al., 2023).

To address these issues, we propose a novel KV cache pruning method, named SlimKV, which ensure the retention of large activations within the KV cache while supporting both Multi-Head Attention (MHA) and Grouped-Query Attention (GQA) architectures. Specifically, we introduce a new KV cache pruning metric that identifies large activations in both Key and Value cache and calculate the

importance of KV cache within each group, preserving these important KV cache during inference to maintain optimal model performance.

We conducted extensive experiments on various long-context LLM generation tasks using popular models with different attention architectures, including MHA and GQA. The experimental results demonstrate that our method not only reduces the storage requirements of the KV cache but also maintains superior model performance. Specifically, It demonstrates similar accuracy while utilizing only 1/10 of the KV cache compared to existing SOTA methods.

**Contributions**:

- This paper addresses the importance of identifying and retaining large activations when prune KV cache. It further examines the limitations of existing KV cache pruning methods, which fail to account for large activations in Value cache and do not support GQA.

- We propose a novel KV pruning method that supports both MHA and GQA architectures while retaining large activations in both Key and Value cache, thereby enhancing overall performance.

- Experimental results on various long-context datasets show that our method achieves comparable accuracy to baseline methods while reducing the required only 1/10 KV cache.

## 2 BACKGROUND & MOTIVATION

### 2.1 GROUPED-QUERY ATTENTION AND KV CACHE

In the context of LLMs and during the prefill phase of the inference process, the attention mechanism within a specific attention group $g$ is described by the following computational steps for the $h$-th attention head in $g$:

$$A^h = \text{Softmax}\left(\frac{Q^h(K^g)^T}{\sqrt{d_k}}\right)$$
$$O^h = A^h V^g \tag{1}$$

Here, $Q^h$, $K^g$, and $V^g$ stand for the query, key, and value vectors, respectively, which are fundamental components of the attention layer. The attention weights $A^h$ are calculated by taking the dot product of the query vectors $Q^h$ with the transposed key vectors $K^g$, scaling the result by the inverse square root of the dimensionality of the key vectors ($\sqrt{d_k}$), and then applying the softmax function to ensure the weights sum to one. The final output for the attention head $O^h$ is produced by weighting the value vectors $V^g$ by the computed attention weights $A^h$. MHA can be regarded as a special case of GQA, wherein each group contains exactly one query head.

### 2.2 IMPORTANCE OF LARGE ACTIVATIONS IN KV CACHE

Large activations refer to the neuron outputs in LLMs that have high absolute values. Recent studies (Sun et al., 2024) have indicated that removing these large activations in LLMs can lead to a significant reduction in model performance, thus sustain large activations in KV cache is essential for maintaining model performance during pruning. The significance of large activations within the KV cache during the pruning process is profound, primarily due to their substantial impact on the outputs of models. These large activations, present in both the Key and Value cache, wield considerable influence over the resulting computational outcomes, and their exclusion can precipitate significant performance degradations in models. We delve into how these large activations crucially influence both the Key and Value cache.

**Large Activations of Key Cache**. Key cache is pivotal in shaping the distribution of attention logits. Within the architecture of self-attention, Key cache interact with query vectors to compute attention logits. When one specific Key cache is important, it profoundly affects the softmax computation, resulting in elevated attention logits at specific positions. This increased attention indicates that the associated value vectors, which are weighted according to these attention logits, will exert a stronger influence on the resultant output. Consequently, pruning large activations of Key cache can impair the

model's capacity to concentrate on essential aspects of the input data, potentially leading to significant information loss and a decrement in model efficacy.

**Large Activations of Value Cache**. Similarly, large activations in the Value cache directly determine the magnitude of output components. The value vectors, when augmented by their corresponding attention logits, make a direct contribution to the final output. Thus, large activations in Vaue cache lead to more substantial contributions to output, particularly when aligned with high attention logits. These vital elements of Value cache contain essential information crucial for the accuracy and richness of the model's output. Neglecting these large activations during pruning could result in the loss of critical output features, thereby diminishing the model's overall effectiveness and accuracy.

## 2.3 Limitations of Existing KV Cache Pruning Methods

Current pruning methods for KV cache primarily focus on Key cache and tend to overlook Value cache that could impact model performance. Here, we explore the inherent limitations of these methods.

**Neglect of Large Activations in the Value Cache**. Existing approaches to KV cache pruning typically utilize large activations in the attention logits as metric. This methodology, however, often overlooks large activations in the Value cache. The Value cache is feature of input data, and when this data, weighted by attention logits, significantly influences the output, its importance cannot be understated. By not considering large activations in Value cache, current methods might discard valuable information, potentially leading to a substantial decline in the accuracy and effectiveness of the model's output.

**Incompatibility with Grouped-Query Attention**. Traditional pruning methods generally prune KV cache on a per-head basis, which can lead to issues when dealing with architectures like GQA, where queries within a group share a common KV cache. This shared mechanism in GQA is designed to optimize processing efficiency and coherence across grouped queries, but it poses a challenge for typical pruning strategies. Since each head's KV cache is independently assessed and pruned in these traditional methods, they fail to accommodate the shared nature of KV caches in GQA settings, leading to potential inefficiencies and inconsistencies in how information is retained or discarded across groups.

## 3 Design of SlimKV

The core objective of SlimKV is to minimize the usage of the KV cache while ensuring the preservation of model performance as much as possible. This necessitates identifying which KV cache are important and retaining them. We hypothesize that not only the large activations of Key cache determine the importance of KV cache but also the large activations in Value cache play a significant role. Hence, a comprehensive metric needs to be designed to assess the importance of specific KV cache.

## 3.1 Grouped-Wise Recognition of Large Activations of Key Cache

To effectively identify Key cache with large activations, we employ a methodology that traces back from large attention logits to their corresponding significant Key cache. Specifically, regions with elevated attention logit values indicate a proportional impact on the model's output, thereby warranting the preservation of the Key cache associated with these logits. Consequently, our system gives precedence to maintaining KV cache linked to these high-magnitude activations within the attention logits to ensure that essential contextual connections are retained during the pruning process. Moreover, to accommodate both GQA and MHA architectures, we introduce a grouped-wise voting mechanism to facilitate the recognition of large activations. By doing so, we curtail the quantity of KV caches during the generation phase, thus alleviating the computational load for LLMs when processing extensive contexts. To delineate our methodology with precision, we propose the subsequent definitions.

- **Prompt Length** ($L_{\text{prompt}}$): The total length of the input provided by the user.

- **Observation Window** ($L_{\text{obs}}$): This refers to the most recent segment of the prompt that is critical for analyzing how various contexts affect attention distribution.

- **Prefix Length** ($L_{\text{prefix}}$): This is the length of the input that precedes the observation window. It is included in the prompt but does not overlap with the observation window. The relationship between these lengths is given by:

$$L_{\text{prompt}} = L_{\text{prefix}} + L_{\text{obs}} \tag{2}$$

Finally, we obtain the grouped-wise importance score of Key cache. This process involves computing the attention logits for each query within the observation window across all query heads in one attention group. These weights are then aggregated to identify the most significant positions in the prefix. For a single batch sequence, the computation can be formally expressed as:

$$\mathbf{S_1} = \sum_{h=0}^{N} \sum_{i=0}^{L_{\text{obs}}} \mathbf{W}_{\text{obs}}[h, i, :] \tag{3}$$

Here, tensor $\mathbf{S_1} \in \mathbb{R}^{L_{\text{prefix}}}$ represents the importance of Key cache within current attention group. The tensor $\mathbf{W}_{\text{obs}} \in \mathbb{R}^{N \times L_{\text{obs}} \times L_{\text{prefix}}}$ represents the subset of softmax-normalized attention logits over $N$ heads in current attention group.

### 3.2 Recognition of Large Activations in Value Cache

To identify important information in the Value cache, the most direct and effective strategy is to focus on large activations, which are typically indicative of key components in the model's output. These large activations in the Value cache directly affect the magnitude of the output. The importance of Value cache based on these large activations can be computed as follows:

$$\mathbf{S_2} = \max(|V|, \dim = -1) \tag{4}$$

where $\mathbf{S_2} \in \mathbb{R}^{L_{\text{prefix}}}$ represents the importance of the Value cache, and the $\max$ operation extracts the maximum magnitude from each Value cache. Here, $\mathbf{V} \in \mathbb{R}^{L_{\text{prefix}} \times d_K}$ denotes the Value matrix.

To comprehensively assess the importance of various KV caches, we propose an integrated metric that combines the indicators $S_1$ and $S_2$, which respectively reflect the importance of the Key and Value cache. The overall importance of one specific KV cache is thus quantified by the product of these two indicators:

$$\mathbf{S} = \mathbf{S_1} * \mathbf{S_2} \tag{5}$$

This metric allows us to capture the joint impact of both the Key and Value components on the significance of the KV cache within the system.

### 3.3 Implementation of SlimKV

The core approach of SlimKV involves identifying and selecting the most crucial KV cache per attention group. Listing 1 shows the PyTorch-style pseudo code of SlimKV. Overall, SlimKV operates through two stages as follows:

- **Scoring for KV cache.** By the scoring process defined above (Eq. 5), we select the important KV cache based on the observation window. We also highlight the consistency of the attention logits pattern within observation windows throughout the generation, suggesting that these selected KV cache are also vital for subsequent generation. Furthermore, we implement clustering to retain the KV cache surrounding the selected KV cache. Line 8-21 shows the pseudo code of the scoring process for KV cache.

- **Update and store compressed KV cache.** We concatenate the selected KV cache with all KV cache within the observation window, which encompasses all the necessary prompt information. Line 22-28 shows the compressing process. The concatenated KVs are stored for later use in generation, thereby saving memory usage.

```python
def slimkv(query_states, key_states, value_states, window_size, max_capacity_prompt,
     kernel_size):
    bsz, num_heads, q_len, head_dim = query_states.shape
    # Ensure it is the prompt phase.
    assert key_states.shape[-2] == query_states.shape[-2]
    if q_len < max_capacity_prompt:
        return key_states, value_states
    else:
        # Compute attention weights of observing window's queries and prefix context's Keys.
        attn_weights = compute_attn(query_states[..., -window_size:, :], key_states,
     attention_mask)
        # Sum the weight along the query dimension to obtain S1.
        S1 = attn_weights[..., -window_size:, :-window_size].sum(dim=-2)
        # Calculate S2 for prefix KV cache
        S2 = torch.max(torch.abs(value_states[..., :, :-window_size]), dim=-1)
        # Combine S1 and S2 for final KV cache metric
        S = S1 * S2
        # Apply 1D pooling for clustering.
        pool_S = pool1d(S, kernel_size=kernel_size, padding=kernel_size//2, stride=1)
        # Select top-k indices based on the pooled weights to identify important positions.
        indices = pool_S.topk(max_capacity_prompt - window_size, dim=-1).indices
        # Expand the indices to match the head dimension for gathering.
        indices = indices.unsqueeze(-1).expand(-1, -1, -1, head_dim)
        # Gather the compressed past key and value states based on the selected indices.
        k_past_compress = key_states[..., :-window_size, :].gather(dim=2, index=indices)
        v_past_compress = value_states[..., :-window_size, :].gather(dim=2, index=indices)
        k_obs = key_states[..., -window_size:, :]
        v_obs = value_states[..., -window_size:, :]
        key_states = torch.cat([k_past_compress, k_obs], dim=2)
        value_states = torch.cat([v_past_compress, v_obs], dim=2)
        return key_states, value_states
```

Listing 1: Implementation of SlimKV in pseudo PyTorch style.

## 4 EVALUATION

We conducted extensive experiments to validate two primary questions: 1) SlimKV outperforms other KV cache pruning methods in maintaining model performance after preserving large activations; 2) SlimKV effectively supports the GQA model while reducing the GPU memory footprint of LLMs' KV cache. Firstly, we introduce the underlying LLMs (Section 4.1), the datasets used for evaluation (Section 4.2), and the baseline methods for comparison (Section 4.3). Subsequently, we present the performance of SlimKV in memory-oriented scenarios with MHA models (Section 4.4). Following this, we report on the capability of SlimKV to support GQA models (Section 4.5). Finally, we discuss the effectiveness of the different components that constitute SlimKV (Section 4.6).

### 4.1 BACKBONE LLMS

We compare SlimKV against baselines using state-of-the-art open-sourced LLMs, namely Llama-3-8B (Dubey et al., 2024), Llama-2-7B (Touvron et al., 2023), Qwen-2-7B (Yang et al., 2024), Mistral-7B (Jiang et al., 2023). Testing examples are evaluated in a generative format, with answers generated by greedy decoding across all tasks to ensure a fair comparison.

### 4.2 DATASETS

We use LongBench (Bai et al., 2024) to assess the performance of SlimKV on tasks involving long-context inputs. LongBench is a meticulously designed benchmark suite that tests the capabilities of language models in handling extended documents and complex information sequences. This benchmark was created for multi-task evaluation of long context inputs. It includes 17 datasets covering tasks such as single-document QA (Kočiský et al., 2018; Dasigi et al., 2021), multi-document QA (Yang et al., 2018; Ho et al., 2020), summarization (Huang et al., 2021; Zhong et al., 2021; Fabbri et al., 2019), few-shot learning (Li and Roth, 2002; Gliwa et al., 2019; Joshi et al., 2017), synthetic, and code generation (Guo et al., 2023; Liu et al., 2023). The datasets feature an average input length ranging from 1,235 to 18,409 tokens (detailed average lengths can be found in Table 2), necessitating substantial memory for KV cache management. For all these tasks, we adhered to the standard metrics recommended by LongBench (i.e., F1 for QA, Rouge-L for summarization, Acc for synthetic and Edit Sim for code generation.) We refer readers to more details at Appendix A.

| Method | Single-Document QA | | | Multi-Document QA | | | Summarization | | | Few-shot Learning | | | Synthetic | | Code | | Avg. |
|---|---|---|---|---|---|---|---|---|---|---|---|---|---|---|---|---|---|
| | NrtvQA | Qasper | MF-en | HotpotQA | 2WikiMQA | Musique | GovReport | QMSum | MultiNews | TREC | TriviaQA | SAMSum | PCount | PRe | Lcc | RB-P | |
| | 18409 | 3619 | 4559 | 9151 | 4887 | 11214 | 8734 | 10614 | 2113 | 5177 | 8209 | 6258 | 11141 | 9289 | 1235 | 4206 | |
| Llama-2-7b, KV Size = Full | | | | | | | | | | | | | | | | | |
| FKV | 18.39 | 21.05 | 35.54 | 31.69 | 25.38 | 10.14 | 24.77 | 20.92 | 2.33 | 64.0 | 83.38 | 40.99 | 5.5 | 9.5 | 59.31 | 52.85 | 31.61 |
| Llama-2-7b, KV Size = 256 | | | | | | | | | | | | | | | | | |
| SKV | 14.68 | 18.47 | 30.35 | 31.49 | 25.13 | 8.79 | 17.26 | 20.18 | 1.85 | 56.5 | 83.31 | 37.85 | 6.0 | 9.0 | 55.9 | 50.74 | 29.22 |
| H2O | 15.47 | 16.96 | 27.82 | 30.84 | 25.12 | 7.91 | 14.47 | 19.65 | 1.44 | 42.0 | 79.93 | 38.17 | 5.0 | 8.5 | 56.12 | 51.02 | 27.53 |
| Ours | 14.91 | 19.1 | 31.49 | 31.57 | 24.91 | 9.07 | 17.25 | 20.43 | 1.9 | 58.5 | 84.12 | 38.88 | 6.0 | 10.5 | 56.79 | 51.03 | **29.78** |
| Llama-2-7b, KV Size = 2048 | | | | | | | | | | | | | | | | | |
| SKV | 18.01 | 21.86 | 35.65 | 31.72 | 25.42 | 9.86 | 23.34 | 20.78 | 2.26 | 64.0 | 83.46 | 40.92 | 5.5 | 9.0 | 58.99 | 51.95 | 31.42 |
| H2O | 18.19 | 21.48 | 34.39 | 31.32 | 26.4 | 9.71 | 22.29 | 21.07 | 2.28 | 63.0 | 82.96 | 41.0 | 5.5 | 8.5 | 58.46 | 52.07 | 31.16 |
| Ours | 18.1 | 21.51 | 35.68 | 31.76 | 25.42 | 9.85 | 23.56 | 20.63 | 2.37 | 64.0 | 83.38 | 40.44 | 5.5 | 9.0 | 59.25 | 53.82 | **31.52** |

Table 1: The retention of large activations during KV cache pruning can substantially preserve model performance. We present a performance comparison of SlimKV (Ours) with SnapKV (SKV), H2O, and FullKV (FKV) on the LongBench dataset for Multi-Head Attention (MHA) LLMs. Generally, SlimKV outperforms competing KV cache compression methods across a spectrum of KV cache budget sizes. The superior performance of SlimKV becomes particularly pronounced at smaller KV cache size (e.g., KV Size = 256). Text in bold denotes the highest performance metrics achieved.

## 4.3 BASELINES

We compare SlimKV with robust baseline methods, including **StreamingLLM (SLM)** (Xiao et al., 2024), **Heavy Hitter Oracle (H2O)** (Zhang et al., 2023), **SnapKV (SKV)** (Li et al., 2024) and **FullKV (FKV)**. All of which maintain a consistent KV cache size budget across various layers, albeit employing different strategies for the selection of important KV cache. It is important to note that while H2O and SKV are exclusively applicable to MHA models, the remaining methods can be utilized in both MHA and GQA models.

## 4.4 EFFECTIVENESS OF PRESERVING LARGE ACTIVATIONS

In our experimental evaluation on the Multi-Head Attention (MHA) model, we compare the performance of our proposed method, SlimKV, with other methods that do not preserve large activations during KV cache pruning. The assessment results, sourced from the LongBench dataset (Bai et al., 2024) using the Llama-2-7B model, are delineated in Table 1. This table presents the outcomes for two distinct KV cache dimensions: 256 and 2048. These dimensions epitomize two divergent operational conditions: one that is memory-constrained and another that prioritizes performance retention, thus underscoring the inherent trade-off between memory conservation and model efficacy.

In a comprehensive analysis, SlimKV is observed to consistently outperform other contemporary state-of-the-art (SOTA) pruning methods in terms of maintaining model performance. With a KV cache size budget constrained to 256, SlimKV secures an average accuracy of 29.78, which is notably higher than the 29.22 average accuracy achieved by other SOTA techniques. In scenarios where the KV cache size budget is expanded to 2048, SlimKV demonstrates an average accuracy of 31.52, marginally surpassing the 31.42 average accuracy recorded by its SOTA counterparts. These findings emphasize the critical role of large activation preservation in sustaining the long-contextual modeling proficiency of expansive neural models.

Moreover, SlimKV exhibits exceptional performance across a diverse array of sub-tasks. For instance, when operating with a KV cache size budget of 256, SlimKV outstrips competing approaches in terms of accuracy on Few-shot Learning tasks. This indicates that the model adeptly consolidates information from a limited number of examples, thereby accentuating the prospects for in-depth exploration into in-context learning paradigms.

## 4.5 SUPPORT FOR GQA ARCHITECTURE

In this subsection, we focus on showcasing the comparative performance of SlimKV and other KV cache pruning methods across various Grouped-Query Attention LLMs. The evaluation results, drawn from the LongBench dataset (Bai et al., 2024) and encompassing models such as Llama-3-8B, Mistral-

| Method | Single-Document QA | | | Multi-Document QA | | | Summarization | | | Few-shot Learning | | | Synthetic | | Code | | Avg. |
|---|---|---|---|---|---|---|---|---|---|---|---|---|---|---|---|---|---|
| | NrtvQA | Qasper | MF-en | HotpotQA | 2WikiMQA | Musique | GovReport | QMSum | MultiNews | TREC | TriviaQA | SAMSum | PCount | PRe | Lcc | RB-P | |
| | 18409 | 3619 | 4559 | 9151 | 4887 | 11214 | 8734 | 10614 | 2113 | 5177 | 8209 | 6258 | 11141 | 9289 | 1235 | 4206 | |
| LlaMa-3-8B-Instruct, KV Size = Full | | | | | | | | | | | | | | | | | |
| FKV | 25.70 | 29.75 | 41.12 | 45.55 | 35.87 | 22.35 | 25.63 | 23.03 | 26.21 | 73.00 | 90.56 | 41.88 | 4.67 | 69.25 | 58.05 | 50.77 | 41.46 |
| LlaMa-3-8B-Instruct, KV Size = 128 | | | | | | | | | | | | | | | | | |
| SLM | 18.61 | 9.65 | 25.99 | 37.95 | 29.39 | 16.34 | 18.03 | 20.11 | 20.08 | 43.50 | 74.08 | 29.86 | 5.90 | 69.50 | 47.47 | 45.60 | 32.00 |
| Ours | 20.05 | 8.31 | 31.22 | 40.72 | 29.91 | 18.1 | 16.82 | 20.92 | 18.59 | 44.5 | 89.36 | 37.7 | 5.43 | 69.50 | 54.74 | 51.59 | **34.84** |
| LlaMa-3-8B-Instruct, KV Size = 2048 | | | | | | | | | | | | | | | | | |
| SLM | 21.71 | 25.78 | 38.13 | 40.12 | 32.01 | 16.86 | 23.14 | 22.64 | 26.48 | 70.00 | 83.22 | 31.75 | 5.74 | 68.50 | 53.50 | 45.58 | 37.82 |
| Ours | 25.79 | 29.08 | 41.15 | 45.26 | 34.62 | 22.38 | 25.27 | 22.87 | 26.44 | 72.5 | 90.56 | 41.22 | 5.1 | 68.75 | 58.39 | 52.27 | **41.35** |
| Mistral-7B-Instruct, KV Size = Full | | | | | | | | | | | | | | | | | |
| FKV | 25.43 | 31.72 | 48.31 | 42.21 | 26.78 | 17.50 | 25.45 | 23.90 | 4.98 | 68.50 | 86.33 | 42.44 | 5.00 | 88.40 | 51.30 | 47.89 | 39.76 |
| Mistral-7B-Instruct, KV Size = 128 | | | | | | | | | | | | | | | | | |
| SLM | 16.57 | 14.68 | 32.40 | 30.19 | 22.64 | 12.34 | 18.08 | 18.96 | 3.71 | 43.50 | 74.22 | 29.02 | 4.50 | 29.48 | 39.23 | 36.16 | 26.60 |
| Ours | 20.66 | 17.64 | 40.78 | 36.21 | 23.17 | 14.06 | 15.81 | 21.1 | 3.54 | 45.0 | 84.23 | 39.19 | 4.5 | 35.69 | 45.57 | 42.15 | **30.58** |
| Mistral-7B-Instruct, KV Size = 2048 | | | | | | | | | | | | | | | | | |
| SLM | 20.31 | 26.64 | 45.72 | 35.25 | 24.31 | 12.20 | 27.47 | 21.57 | 4.87 | 68.50 | 71.95 | 31.19 | 5.00 | 22.56 | 43.38 | 37.08 | 31.12 |
| Ours | 26.2 | 31.8 | 49.2 | 42.5 | 26.8 | 18.11 | 24.88 | 23.22 | 4.85 | 69.5 | 86.06 | 42.58 | 5.5 | 88.4 | 51.74 | 48.14 | **39.97** |
| Qwen-2-7B-Instruct, KV Size = Full | | | | | | | | | | | | | | | | | |
| FKV | 25.11 | 42.64 | 44.46 | 55.02 | 54.66 | 35.96 | 36.18 | 23.43 | 26.53 | 77.0 | 89.99 | 44.88 | 6.75 | 75.92 | 55.73 | 49.03 | 46.46 |
| Qwen-2-7B-Instruct, KV Size = 128 | | | | | | | | | | | | | | | | | |
| SLM | 19.01 | 24.48 | 28.21 | 43.09 | 44.72 | 26.36 | 18.3 | 18.96 | 18.22 | 46.5 | 77.6 | 31.8 | 8.25 | 15.5 | 36.94 | 32.28 | 30.64 |
| Ours | 22.2 | 26.64 | 37.82 | 48.77 | 47.65 | 30.44 | 16.4 | 20.12 | 16.22 | 41.0 | 88.16 | 41.19 | 5.05 | 53.68 | 46.01 | 39.18 | **36.28** |
| Qwen-2-7B-Instruct, KV Size = 2048 | | | | | | | | | | | | | | | | | |
| SLM | 20.26 | 36.26 | 44.17 | 46.56 | 51.83 | 27.67 | 30.46 | 20.69 | 26.73 | 73.5 | 77.12 | 32.95 | 6.2 | 21.0 | 48.95 | 36.1 | 37.53 |
| Ours | 24.34 | 44.01 | 44.81 | 54.02 | 54.46 | 34.65 | 29.73 | 22.8 | 26.23 | 75.5 | 89.84 | 44.22 | 5.75 | 74.0 | 55.09 | 47.48 | **45.43** |

Table 2: SlimKV not only facilitates KV cache pruning for GQA models but also significantly outperforms other pruning methods. We present a performance comparison of SlimKV (Ours) with StreamingLLM (SLM) and FullKV (FKV) on the LongBench dataset for LlaMa-3-8B-Instruct, Qwen-2-7B-Instruct, and Mistral-7B-Instruct models. Across a range of KV cache budget sizes, SlimKV consistently outshines other KV cache compression techniques. Instances of the best performance are highlighted in bold within the text.

7B, and Qwen-2-7B, are presented in Table 2. Within Table 2, we delineate the outcomes for two KV cache sizes: 128 and 2048. These sizes are emblematic of two distinct operational scenarios—the memory-efficient scenario and the performance-preserving scenario, respectively—thus illustrating the balance between memory utilization and model performance.

Overall, SlimKV not only supports KV cache pruning within GQA model architectures but also significantly outperforms existing methods in terms of performance. Across an array of GQA LLMs, SlimKV surpasses other pruning methods: for example, on the Mistral-7B model with a KV cache budget size of 128, SlimKV achieves an average accuracy of 30.58, exceeding the 26.60 average accuracy of other state-of-the-art (SOTA) methods. This advantage is even more pronounced when the KV cache budget size is 2048, where SlimKV attains an average accuracy of 39.97, substantially higher than the 31.12 average accuracy of other SOTA methods.

In scenarios where the KV cache size budget budget is 2048, SlimKV's performance on Mistral-7B even surpasses that of FullKV—meaning that the model, post-SlimKV pruning, sees an average accuracy increase from 39.76 to 39.97. This level of performance enhancement is unattainable by other KV cache pruning methods. Such a phenomenon can be attributed to the presence of certain 'noisy' contexts that, when included, may actually degrade model performance. By excising the KV caches corresponding to these disruptive contexts, one can effectively augment the model's performance.

### 4.6 ABLATION STUDY

To investigate the individual impacts of large activations within the Key and Value caches on model performance, we conducted separate pruning using the metrics $S_1$ from Equation 3 for the Key cache and $S_2$ from Equation 4 for the Value cache. Specifically, $S_1$ was employed to identify and preserve

| Method | Single-Document QA | | | Multi-Document QA | | | Summarization | | | Few-shot Learning | | | Synthetic | | Code | | Avg. |
|---|---|---|---|---|---|---|---|---|---|---|---|---|---|---|---|---|---|
| | NrtvQA | Qasper | MF-en | HotpotQA | 2WikiMQA | Musique | GovReport | QMSum | MultiNews | TREC | TriviaQA | SAMSum | PCount | PRe | Lcc | RB-P | |
| | 18409 | 3619 | 4559 | 9151 | 4887 | 11214 | 8734 | 10614 | 2113 | 5177 | 8209 | 6258 | 11141 | 9289 | 1235 | 4206 | |
| Llama-2-7b, KV Size = Full | | | | | | | | | | | | | | | | | |
| FKV | 18.39 | 21.05 | 35.54 | 31.69 | 25.38 | 10.14 | 24.77 | 20.92 | 2.33 | 64.0 | 83.38 | 40.99 | 5.5 | 9.5 | 59.31 | 52.85 | 31.61 |
| Llama-2-7b, KV Size = 256 | | | | | | | | | | | | | | | | | |
| Ours (only $S_1$) | 15.47 | 16.96 | 27.82 | 30.84 | 25.12 | 7.91 | 14.47 | 19.65 | 1.44 | 42.0 | 79.93 | 38.17 | 5.0 | 8.5 | 56.12 | 51.02 | 27.53 |
| Ours (only $S_2$) | 14.87 | 17.13 | 26.5 | 30.22 | 25.5 | 8.3 | 14.09 | 19.59 | 1.63 | 42.5 | 80.55 | 37.11 | 5.0 | 7.5 | 56.39 | 50.19 | 27.32 |
| Ours (Origin) | 14.91 | 19.1 | 31.49 | 31.57 | 24.91 | 9.07 | 17.25 | 20.43 | 1.9 | 58.5 | 84.12 | 38.88 | 6.0 | 10.5 | 56.79 | 51.03 | **29.78** |

Table 3: Ablation Study: Our findings indicate that focusing exclusively on either the Key cache or the Value cache for large activations leads to a decline in model performance. Empirical results demonstrate that simultaneously addressing large activations in both the Key and Value caches is more effective in preserving the integrity of the model's performance.

large activations in the Key cache, while $S_2$ was utilized for the Value cache. We evaluated the efficacy of SlimKV in pruning on the Llama-2-7B model with a given KV cache budget size of 128. The experimental outcomes are displayed in Table 3.

Overall, focusing solely on large activations within either the Key or Value cache results in a significant decrease in model performance post-KV cache pruning. Specifically, the removal of $S_2$ led to a decline in SlimKV's performance from 29.78 to 27.53, while the removal of $S_1$ resulted in a decrease from 29.78 to 27.32. This underscores that large activations in both the Key and Value caches are crucial for sustaining the model's performance.

## 5 RELATED WORK

**Efficient Attention.** Efficient attention mechanisms have been developed to address the computational and memory inefficiencies of standard self-attention in LLMs. One such approach is the Grouped-Query Attention (GQA), which organizes queries into groups before computing the attention, significantly reducing the number of attention calculations required (Ainslie et al., 2023). Another variant is Multi-Query Attention (MQA), which extends the idea by allowing multiple queries to interact within the same attention framework (Shazeer, 2019). TopK Attention is an adaptive mechanism that focuses only on the top-K most relevant keys for each query, reducing the computational complexity from quadratic to linear with respect to the sequence length. This method not only improves efficiency but also enhances model interpretability by focusing on the most significant interactions (Gupta et al., 2021). Among the linear transformers, Mamba stands out by utilizing a moving-average-based approach to bypass the need for the softmax calculation, thereby linearizing the computational cost with respect to the input length. This model achieves comparable or even superior performance to traditional attention models on several benchmark tasks (Gu and Dao, 2024).

**KV Cache Compression.** Numerous previous works have sought to compress the KV cache by selectively dropping KV pairs using various algorithms. StreamLLM (Xiao et al., 2024), for instance, retains only the most recent tokens and attention sinks (the first few tokens), reducing the KV cache size but potentially discarding important information carried by the middle tokens. The Heavy-Hitter Oracle (H2O) (Zhang et al., 2023) introduces a policy that greedily drops KV pairs during generation based on a scoring function derived from cumulative attention. Similarly, Adaptive KV Compression (FastGen) (Ge et al., 2023) employs a dual-phase algorithm that encompasses four KV cache compression policies. It initially identifies optimal policies through profiling results obtained from prompt encoding, then dynamically evicts caches during the generation phase based on these policies. ScissorHands (Liu et al., 2024) aims to identify and retain pivotal tokens that exhibit a consistent attention weight pattern with previous token windows during generation steps. However, these methods overlook the impact of the Value cache, focusing solely on the Key cache to retain the most important KV pairs. Moreover, they are not applicable to efficient attention architectures like Grouped-Query Attention and Multi-Query Attention, which are utilized by popular LLMs such as Llama-3, Qwen-2, Mistral, Falcon, and others. The method proposed in this paper, referred to as SlimKV, is capable of supporting efficient attention architectures while fully considering the preservation of important information in the KV cache.

## 6 CONCLUSION

This paper presents a novel approach to KV cache pruning for Large Language Models (LLMs) that addresses the critical challenge of efficiently managing increasing context lengths. Our proposed method, SlimKV, effectively retains large activations in both Key and Value caches, thereby enhancing model performance while supporting both Multi-Head Attention (MHA) and Grouped-Query Attention (GQA) architectures.

Through extensive experimentation, we demonstrate that our method reduces more GPU memory usage by than baseline methods, while maintaining comparable accuracy. This work not only highlights the significance of preserving large activations during KV cache pruning but also contributes to the ongoing efforts to optimize LLM serving, ensuring that these powerful models can be utilized more efficiently in practical applications. Future research can build upon our findings to further enhance the efficiency and performance of LLMs in diverse contexts.

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

# A    DETAILS OF EVALUATION

We use LongBench (Bai et al., 2024) to assess the performance of SlimKV on tasks involving long-context inputs. LongBench is a meticulously designed benchmark suite that tests the capabilities of language models in handling extended documents and complex information sequences. This benchmark was created for multi-task evaluation of long context inputs. We present the details of metrics, language and data for LongBench at Table 4.

We run all the experiments on NVIDIA V100 and A100.

| Dataset | Source | Avg len | Metric | Language | #data |
|---|---|---|---|---|---|
| Single-Document QA | | | | | |
| NarrativeQA | Literature, Film | 18,409 | F1 | English | 200 |
| Qasper | Science | 3,619 | F1 | English | 200 |
| MultiFieldQA-en | Multi-field | 4,559 | F1 | English | 150 |
| Multi-Document QA | | | | | |
| HotpotQA | Wikipedia | 9,151 | F1 | English | 200 |
| 2WikiMultihopQA | Wikipedia | 4,887 | F1 | English | 200 |
| MuSiQue | Wikipedia | 11,214 | F1 | English | 200 |
| Summarization | | | | | |
| GovReport | Government report | 8,734 | Rouge-L | English | 200 |
| QMSum | Meeting | 10,614 | Rouge-L | English | 200 |
| MultiNews | News | 2,113 | Rouge-L | English | 200 |
| Few-shot Learning | | | | | |
| TREC | Web question | 5,177 | Accuracy (CLS) | English | 200 |
| TriviaQA | Wikipedia, Web | 8,209 | F1 | English | 200 |
| SAMSum | Dialogue | 6,258 | Rouge-L | English | 200 |
| Synthetic Task | | | | | |
| PassageCount | Wikipedia | 11,141 | Accuracy (EM) | English | 200 |
| PassageRetrieval-en | Wikipedia | 9,289 | Accuracy (EM) | English | 200 |
| Code Completion | | | | | |
| LCC | Github | 1,235 | Edit Sim | Python/C#/Java | 500 |
| RepoBench-P | Github repository | 4,206 | Edit Sim | Python/Java | 500 |

Table 4: An overview of the dataset statistics in LongBench (Bai et al., 2024). 'Source' denotes the origin of the context. 'Accuracy (CLS)' refers to classification accuracy, while 'Accuracy (EM)' refers to exact match accuracy.

