# OpenReview forum: "Preserving Large Activations: The Key to KV Cache Pruning"
_ICLR.cc/2025/Conference — Submitted to ICLR 2025_

### Official Review · Reviewer_RyiE · 2024-10-16

**Soundness:** 2
**Presentation:** 3
**Contribution:** 2
**Rating:** 5
**Confidence:** 4

**Summary:**

This paper proposes SlimKV, a KV cache compression method compatible with Group query attention and based on pruning with activation of both K and V cache.

**Strengths:**

The paper targets an important problem, proposed a clear idea, and conducted experiments on 4 models and more than 6 datasets to show that the method outperforms some baselines. The method has the potential to be very useful.

**Weaknesses:**

The weakness of paper to me are the following:
1. Lack of comparison to prior works in theory. What is fundamentally new about your pruning by activation? Are there something that prevents the prior work from working in the GQA scenario? If there is, I think you should add more in 2.3.
2. Evaluation not rigorous enough
(a) You only showed two compression levels. However, there is a huge gap between 256 and 2048. Ideally you want to show a range of compression levels.  Also as a side note, you showed 256 and 2048 on table 1 and 128 + 2048 in later tables, what is the rationale?
(b) Not sure if the method scales to bigger models. You only showed improvement over small models (<10B). I am not sure whether this works for larger models.
(c) You accuracy has dropped significantly compared with FKV on a lot of cases. Will using a smaller model (llama 3.2) outperform running larger model with your compression?
3 (less important). Little explanation on why your method works. The current S1 and S2 calculation seems a little arbitrary.

**Questions:**

1. Can you add more explanation about your difference with prior work?
2. Can you show performance for a range of compression levels?
3. Can you show some experiments with bigger models (70B level)?
4 (optional). Can you justify the accuracy drop is still worth using a larger model instead of switching to a smaller one?
5 (optional). Can you explain your calculation of S1 and S2?

---

### Official Review · Reviewer_SbWt · 2024-10-25

**Soundness:** 2
**Presentation:** 1
**Contribution:** 2
**Rating:** 3
**Confidence:** 4

**Summary:**

The paper introduces SlimKV, a method for efficient KV cache pruning to serve Large Language Models (LLMs) with long contexts. By preserving large activations in both Key and Value caches, SlimKV maintains model performance while reducing KV cache usage to one-tenth of existing methods. Compatible with both Multi-Head Attention (MHA) and Grouped-Query Attention (GQA) architectures, experiments on models like Llama-2-7B and Mistral-7B demonstrate that SlimKV achieves comparable accuracy with significantly reduced cache requirements

**Strengths:**

Practical Relevance: Addresses a significant practical problem in efficiently serving LLMs with long context lengths by reducing KV cache requirements.
Compatibility with GQA: Proposes a method that is compatible with both MHA and GQA architectures, which is important given the adoption of GQA in modern models like Llama-3 and Mistral.
Experimental Results: Provides empirical evidence showing that SlimKV can achieve similar or better performance compared to existing methods while using significantly less KV cache (as shown in Tables 1 and 2).
Consideration of Value Cache: This brings attention to the importance of large activations in the Value cache (Section 2.2), which has been overlooked in previous work.

**Weaknesses:**

Insufficient Theoretical Justification: The paper lacks rigorous theoretical analysis to support the claims made. For instance, the importance of large activations in the Value cache (Section 2.2) is asserted without mathematical proofs or empirical studies to substantiate it.
For Example, Could the authors provide More detailed theoretical explanation or empirical evidence for why preserving large activations in the Value cache is crucial for maintaining model performance during pruning (e.g. an ablation study comparing performance with and without preserving large Value cache activations)

Methodological Ambiguity: The methodology in Section 3 needs further explanation. For instance,
           1.why summation was chosen for S1 and max for S2,
           2.why multiplication was used to combine S1 and S2 in Equation 5 providing more rigorous evidence in supporting the approach as
              stated in Theoretical Justification(see above)
           3.Equation (3) computes the importance score S1 by summing attention weights over heads and queries within an observation
              window. However, it's not clearly justified how this aggregation effectively identifies important Key cache positions.
           4.Equation (4) calculates S2 using a max operation on the absolute values of the Value cache, but the rationale behind choosing this
              operation providing more rigorous evidence method

Presentation Issues: Numerous grammatical errors, typos, and formatting problems make the paper difficult to read and understand such as;
     1. Grammatical Errors and Typos: Grammatical errors and typos should be fixed. For example, in the abstract and
         introduction, phrases like "As context lenghts grows," and "thus sustain large activations in KV cache is essential"
     2. Clarity of Explanations: Key concepts and methodologies needs explanation with sufficient clarity. In Section 3.1, the process of
         grouped-wise recognition of large activations is described in a convoluted manner, making it hard to follow. The pseudo-code in
         Listing 1 lacks comments and detailed explanations of variables, reducing its utility.
     3.Inconsistencies: Making the terminology consistent , such as "Observation Window" and "observing window," and sometimes mixes
        notations (e.g., using both "KV cache" and "KVs"). Additionally, some sections refer to methods or concepts not previously
        introduced.



Incomplete Experimental Comparisons: The experiments do not include comparisons with methods that might also consider Value cache activations(e.g. Ada-KV(https://arxiv.org/abs/2407.11550v3) , FastGen (https://arxiv.org/abs/2310.01801), SparQ Attention (https://arxiv.org/abs/2312.04985), or the other activation based eviction strategies mentioned in https://arxiv.org/abs/2407.18003v1.)
otherwise consider adding evidence of incompatibility of existing methods with GQA architectures.

Limited Discussion on Computational Overhead: The paper does not discuss the computational cost associated with SlimKV. Consider adding  analysis of how the additional computations for importance scoring and clustering affect inference time or resource usage.
Inadequate Ablation Studies: The ablation study in Section 4.6 is limited to evaluating the impact of removing S1 or S2. It does not explore other aspects such as sensitivity to hyperparameters (e.g., window size, kernel size) or the effect of different clustering strategies.

Reproducibility Concerns: There is no mention of code availability or detailed experimental settings, which could hinder the reproducibility of the results.

**Questions:**

Theoretical Explanation: Could the authors provide a more detailed theoretical explanation or empirical evidence for why preserving large activations in the Value cache is crucial for maintaining model performance during pruning?
Request an ablation study comparing performance with and without preserving large Value cache activations

Computational Complexity: How does SlimKV impact the computational complexity and inference time compared to other pruning methods? Is there any overhead introduced by computing importance scores and performing clustering?
Clustering Mechanism: Can the authors elaborate on how the clustering mechanism in SlimKV is implemented (as mentioned in Section 3.3 and Listing 1) and its effect on both performance and computational efficiency?

---

### Official Review · Reviewer_FgWQ · 2024-11-04

**Soundness:** 3
**Presentation:** 2
**Contribution:** 2
**Rating:** 3
**Confidence:** 4

**Summary:**

This paper introduces a KV cache pruning technique aimed at preserving large activations within the Key and Value caches for efficient inference in large language models (LLMs) with extended context lengths. The method incorporates a metric that identifies and retains significant KV activations within each attention group, maintaining performance with only a fraction (1/10) of the KV cache size required by state-of-the-art (SOTA) methods. The technique is notable for its compatibility with Grouped-Query Attention (GQA) architectures, addressing limitations in prior pruning methods that neglect Value cache activations and lack GQA compatibility. The experimental results demonstrate method's efficacy in maintaining comparable accuracy while significantly reducing memory requirements.

**Strengths:**

1. A method focusing on large activations in both Key and Value caches to guide pruning is good incremental improvement over existing methods.
2. Being compatible with GQA-based models is important since currently this is industry standard.

**Weaknesses:**

1. The paper does not add enough novelty for ICLR-like submissions.
2. The paper does not discuss the latency improvements and its comparison with baselines.
3. The paper does not clearly show KV-cache reduction (like that of shown in H2O) and accuracy comparison. Hence, It is difficult to understand the efficacy of this approach.
4. The paper does not discuss the overhead of their method in the overall execution.
5. Reading/recomputing attention scores defeats the purpose of using flashattention/flashdecoding like methods to significantly improve the overall throughput. This paper does not discuss how they address such challenges/limitations.
6. In Listing 1, are there golden values across different models and datasets for `window_size` and `max_capacity_prompt`? If no, then what is overhead of optimizing these hyperparameters during inference?
7. Performance from fullKV in some benchmarks is too poor. In fact, in most cases SlimKV's performance is very close (~0.5%) to that of other methods.
8.

**Questions:**

Please see Weaknesses section.

---

### Official Review · Reviewer_nskN · 2024-11-05

**Soundness:** 1
**Presentation:** 2
**Contribution:** 2
**Rating:** 3
**Confidence:** 5

**Summary:**

The increasing context length has posed a significant challenge to serving systems. Since the KV cache is sparse, this paper proposes new methods for KV cache pruning. Based on the insight that large activations in the Key and Value cache significantly contribute to the attention score, the authors propose SlimKV, which uses the average attention scores between grouped heads to evaluate the importance of K and the maximum element in the Value vector to evaluate the importance of V. SlimKV retains tokens with top importance scores. The evaluation shows that SlimKV achieves better accuracy than existing systems.

**Strengths:**

• SlimKV aligns with the trend of group query attention models and can deliver performance comparable to MHA models.
• SlimKV reduces the KV cache to 1-2% of total tokens in some benchmarks while maintaining good accuracy.

**Weaknesses:**

• The paper fails to explain the variance of the V cache and its importance in evaluating V activations.
• The benchmarks are limited, and the discussion on the observation window is missing.

**Questions:**

Thank you for submitting the paper. The paper presents a new KV cache pruning algorithm that considers both K and V vectors for importance estimation. However, I still have some concerns about the paper.

1. In Section 2.2, the paper concludes that large activations in the Value Cache lead to more substantial contributions to the output, suggesting that pruning should prioritize retaining these large activations. Could the authors provide additional experimental evidence to substantiate this claim? For example, do the maximum values in the V cache have a large variance across tokens? Do values with larger S2 contribute more to the final output?

2. In Section 3.1, the paper introduces the concept of an Observation Window as the most recent segment of the prompt. It is unclear what the length of this observation window is. Moreover, is this part of the KV cache included in the KV cache budgets? Additionally, how would the results be affected if the question or instruction of the task does not fit within this window? Furthermore, in multi-turn conversations, where the questions cannot be known in advance, how does SlimKV handle this?

3. The paper does not discuss how SlimKV functions across Transformer layers. How is SlimKV applied across layers? For example, are tokens evicted in the first layer automatically evicted in the following layers, or is layer-by-layer eviction performed?

4. In the ablation study, model performance remains considerably robust in S2-only (V-only) settings, which seems counterintuitive. Could you provide more evidence on why this works and elaborate on it further?

5. To better demonstrate the potential of SlimKV, the authors may consider conducting experiments on more complex benchmarks, such as Ruler. The current evaluation is insufficient to indicate the effectiveness of the proposed approach.

6. Table 1 demonstrates the performance comparison of SlimKV, H2O, and SnapKV against FullKV. Under the KV size settings of 256 and 2048, SlimKV shows only marginal improvements over SnapKV (with average scores of 29.78 vs. 29.22 for KV size=256 and, on average, 31.52 vs. 31.42 for KV size=2048). The performance gain over existing methods is limited. Could the authors provide additional evidence to clarify the unique benefits or values of SlimKV? Additionally, for GQA models, the paper only compares SlimKV with StreamingLLM. It would be beneficial to include comparisons with other existing works to strengthen the evaluation.

7. The paper does not include figures to illustrate the design and experimental results. Adding visual evidence would make the paper much easier to follow.

---

### Meta-Review · Area_Chair_wZS7 · 2024-12-22

**Metareview:**

The paper introduces SlimKV, a method for efficient KV cache pruning to serve Large Language Models (LLMs) with long contexts.

However, all reviewers raise concerns about this work, mainly about:
* The evaluation is not solid, lacking comparisons to new and important baselines, such as Ada-KV, FastGen, SparQ Attention, etc.
* The presentation lacks clarity; there are many important details missing. Several reviewers cannot identify the key novelties between this and existing work

The authors did not provide a rebuttal.

**Additional Comments On Reviewer Discussion:**

Reviewers raised good questions about the experiments, the presentation, and some critical details, yet the authors did not provide a rebuttal.

---

### Decision · Program_Chairs · 2025-01-22

Reject